# Generation of Multipotent Stem Cells from Adult Human Peripheral Blood Following the Treatment with Platelet-Derived Mitochondria

**DOI:** 10.3390/cells9061350

**Published:** 2020-05-29

**Authors:** Haibo Yu, Wei Hu, Xiang Song, Yong Zhao

**Affiliations:** Center for Discovery and Innovation, Hackensack Meridian Health, Nutley, NJ 07110, USA; Haibo.Yu@HMH-CDI.org (H.Y.); whu2@stevens.edu (W.H.); Xiang.Song@HMH-CDI.org (X.S.)

**Keywords:** PB-IPC, multipotent stem cells, differentiation, platelets, mitochondria, cell reprogramming

## Abstract

Autologous stem cells are highly preferred for cellular therapy to treat human diseases. Mitochondria are organelles normally located in cytoplasm. Our recent studies demonstrated the differentiation of adult peripheral blood-derived insulin-producing cells (designated PB-IPC) into hematopoietic-like cells after the treatment with platelet-derived mitochondria. To further explore the molecular mechanism and their therapeutic potentials, through confocal and electron microscopy, we found that mitochondria enter cells and directly penetrate the nucleus of PB-IPC after the treatment with platelet-derived mitochondria, where they can produce profound epigenetic changes as demonstrated by RNA-seq and PCR array. Ex vivo functional studies established that mitochondrion-induced PB-IPC (miPB-IPC) can give rise to retinal pigment epithelium (RPE) cells and neuronal cells in the presence of different inducers. Further colony analysis highlighted the multipotent capability of the differentiation of PB-IPC into three-germ layer-derived cells. Therefore, these data indicate a novel function of mitochondria in cellular reprogramming, leading to the generation of autologous multipotent stem cells for clinical applications.

## 1. Introduction

Diabetes is a major public health issue with complex etiology that affects over 350 million people worldwide. Prevalence exceeds 12.1% of the population in India, 11.6% in China [1,2], and 9.3% in the US, and one billion people worldwide are pre-diabetic [3]. Diabetes is the sixth leading cause of death in the U.S., and is associated with increased risk for heart disease, stroke, kidney disease, blindness, and amputations [4,5]. Immune dysfunction is a common factor in both Type 1 Diabetes (T1D) and Type 2 Diabetes (T2D) [6,7,8,9,10]. T1D is characterized by autoimmune destruction of pancreatic islet β cells and disruption of immune cells including T cells, B cells, regulatory T cells (Tregs), monocytes/macrophages (Mo/Mφs), dendritic cells (DCs), natural killer (NK) cells, and natural killer T (NKT) cells [11]. Although T2D is characterized mainly by insulin resistance and aberrant production of insulin, chronic low-grade inflammation in peripheral tissues such as adipose tissue, liver, and muscle also contributes to the disease [6,7,8,12,13,14,15,16]. Specifically, T cells have emerged as unexpected promoters and controllers of insulin resistance [12], by promoting recruitment of inflammatory macrophages to adipose depots and producing inflammatory cytokines that promote the development of insulin resistance leading to diabetes. Despite more than 30 years of intense research, cures for both T1D and T2D remain elusive. 

Deficit of insulin-producing cells is another crucial common issue for T1D and T2D [17,18]. While insulin provides T1D patients the means to manage blood sugar, it is not a cure, and insulin does not address the underlying immune dysfunction that causes β-cell destruction. To overcome the shortage of insulin-producing cells in diabetic patients, pancreas and islet transplantations have offered potential treatments for independence from insulin injections. However, donor scarcity and risk of immune rejection severely hinders their wide applications [19]. To date, insulin-producing cells have been generated from embryonic stem (ES) cells and induced pluripotent stem (iPS) cells through reprogramming by small molecules or by viral transduction of transcription factors [20,21,22]. Thus far, these approaches have been limited by an inability to generate true functional islet β cells in sufficient numbers for clinical use, as well as their safety and ethical concerns and potential immune rejection issues to ES- or iPS derivatives [23,24]. Thus, this compelling need brings a sense of urgency to find a cure for diabetes that can not only halt the progression of autoimmunity in T1D and correct multiple immune dysfunctions in T2D, but also overcome the shortage of insulin-producing β-cells. 

We developed the Stem Cell Educator (SCE) therapy, which harnesses the unique therapeutic potential of cord-blood-derived multipotent stem cells (CB-SC) to treat the multiple immune dysfunctions in T1D [25,26], T2D [27] and other autoimmune disease such as alopecia areata [28]. SCE therapy circulates a patient’s blood through a blood cell separator, co-cultures the patient’s immune cells with adherent CB-SC in vitro, and returns “educated” cells to the patient’s circulation [25,26,27,28]. Our clinical study revealed that the nature of platelets was modulated in diabetic subjects after receiving the treatment with SCE therapy [29]. While exploring the molecular mechanisms underlying the SCE therapy, ex vivo studies demonstrated that platelet-releasing mitochondria can migrate to pancreatic islets and be taken up by human islet β cells, leading to the proliferation and enhancement of islet β-cell functions [29]. To explore the alternative source for islet β cells, we characterized adult peripheral blood insulin-producing cells (PB-IPC) from adult peripheral blood by virtue of their ability to attach to the hydrophobic plastic surface [30], which is a similar approach for an isolation of CB-SC [25,26,27,28,29]. PB-IPC displayed characteristics of islet β-cell progenitors including the expression of β cell-specific insulin gene-associated transcription factors and prohormone convertases, production of insulin, the ability to reduce hyperglycemia and the ability to migrate into pancreatic islets after transplantation into the streptozotocin (STZ)-induced diabetic mice [31]. In our recent study, PB-IPC have been treated with platelet-derived mitochondria to test the potential of improving islet β-cell differentiation. Unexpectedly, our ex vivo and in vivo functional studies confirmed that treatment with platelet-derived mitochondria can reprogram the transformation of adult PB-IPC into functional CD34^+^ hematopoietic stem cells (HSC)-like cells, leading to the production of blood cells such as T cells, B cells, monocytes/macrophages, granulocytes, red blood cells, and megakaryocytes (MK)/platelets (submitted, under review). To further characterize PB-IPC and explore their therapeutic potentials, here we describe a series of findings that indicate mitochondria may be used to transform PB-IPC into multipotent stem cells, giving rise to three-germ layer-derived cells. 

## 2. Materials and Methods

### 2.1. PB-IPC Cell Culture

Human buffy coat blood units (*n* = 42; mean age of 47.64 ± 14.07; age range from 16 to 73 years old; 23 males and 19 females) were purchased from the New York Blood Center (New York, NY, USA, http://nybloodcenter.org/). Human buffy coats were initially added to 40 mL chemical-defined serum-free culture X-VIVO 15^TM^ medium (Lonza, Walkersville, MD, USA) and mixed thoroughly with 10 mL pipette, and then used for isolation of peripheral blood-derived mononuclear cells (PBMC). PBMC were harvested as previously described [32]. Briefly, mononuclear cells were isolated from buffy coats blood using Ficoll-Paque^TM^ PLUS (γ = 1.007, GE Healthcare), followed by removing the red blood cells using Red Blood Cell Lysis buffer (eBioscience, San Diego, CA, USA). After three washes with saline, the whole PBMC were seeded in 150 × 15 mm Petri dishes (BD, Franklin Lakes, NJ, USA) at 1 × 10^6^ cells/mL, 25 mL/dish in chemical-defined serum-free culture X-VIVO 15^TM^ medium (Lonza, Walkersville, MD, USA) without adding any other growth factors and incubated at 37 °C in 8% CO_2_ [33]. Seven days later, PB-IPC were growing and expanded by adhering to the hydrophobic bottom of Petri dishes. Consequently, PB-IPC were washed three times with saline and all floating cells were removed. The serum-free NutriStem^®^ hPSC XF culture medium (Corning, New York, NY, USA) was then added for continued cell culture and expansion at 37 °C in 8% CO_2_. The expanded PB-IPC were usually applied for experiments during 7–14 days. PB-IPC were treated with 100 μg/mL platelet-derived mitochondria for 7–14 days in the non-treated 24-well plates or Petri dishes with the serum-free NutriStem^®^ hPSC XF culture medium (Corning), at 37 °C and 8% CO_2_.

### 2.2. Isolation of Mitochondria from Platelets

The mitochondria were isolated from peripheral blood (PB)-platelets using the Mitochondria Isolation kit (Thermo scientific, Rockford, IL, USA, Prod: 89874) according to the manufacturer’s recommended protocol [29]. Adult human platelet units (*N* = 19) were purchased from the New York Blood Center (New York, NY, USA, http://nybloodcenter.org/). The concentration of mitochondria was determined by the measurement of protein concentration using a NanoDrop 2000 Spectrophotometer (ThermoFisher Scientific, Waltham, MA, USA). The isolated mitochondria were aliquoted and kept in −80 °C freezer for experiments. 

For mitochondrial staining with fluorescent dyes, mitochondria were labeled with MitoTracker Deep Red FM (100 nM) (Thermo Fisher Scientific, Waltham, MA, USA) at 37 °C for 15 min according to the manufacturer’s recommended protocol, followed by two washes with PBS at 3000 rpm × 15 min [29].

### 2.3. Flow Cytometry

Flow cytometric analyses of surface and intra-cellular markers were performed as previously described [29]. PB-IPC were washed with PBS at 2000 rpm for 5 min. Mitochondria were washed with PBS at 12,000 g for 10 min at 4 °C. PB-IPC’s nuclei were washed with PBS at 500 g for 5 min at 4 °C. Samples were pre-incubated with human BD Fc Block (BD Pharmingen, San Jose, CA, USA) for 15 min at room temperature, and then directly aliquoted for different antibody staining. Cells were incubated with different mouse anti-human monoclonal antibodies (mAb) from Beckman Coulter (Brea, CA, USA) including FITC-conjugated anti-CD45RO, anti-CD19, anti-CD4, anti-CD8 and anti-CD42a; phycoerythrin (PE)-conjugated anti-CD34, anti-CCR7 and anti-CXCR4; phycoerythrin-Cy5.5 (PE-Cy5.5)-conjugated anti-CD19, anti-CD117 and anti-SOX2; phycoerythrin-Cy7 (PE-Cy7)-conjugated anti-CD41, anti-CD11b and anti-CD45; APC-conjugated anti-CD34, anti-CXCR4, and anti-CD4; APC-Alexa Fluor 750-conjugated, anti-CD66b and anti-CD8; pacific blue (PB)-conjugated anti-CD38; Krome Orange-conjugated anti-CD14. From BD Biosciences (San Jose, CA, USA), the investigator purchased the FITC-conjugated anti-human lineage cocktail 1 (Lin1) (CD3, CD14, CD16, CD19, CD20, CD56), Alexa Fluor 488-Sox2, Alexa Fluor 647-conjugated mouse anti-human C-peptide and insulin Abs. FITC-conjugated anti-human MAFA ab was obtained from United States Biological (Salem, MA, USA). APC-conjugated mouse anti-human CD36 mAb was purchased from BioLegend (San Diego, CA, USA). PE-conjugated anti-human GLUT2 antibody was purchased from R & D Systems (Minneapolis, MN, USA). Mouse anti SDF-1 polyclonal antibody was purchased from Abcam (Cambridge, MA, USA). The eFluor 660-conjugated rat anti-human OCT3/4 and isotype-matched IgG Abs were from Thermo Fisher Scientific (Waltham, MA, USA). For surface staining, cells were stained for 30 min at room temperature and then washed with PBS at 2000 rpm for 5 min prior to flow analysis. Isotype-matched mouse anti-human IgG antibodies (Beckman Coulter) served as a negative control for all fluorescein-conjugated IgG mAb. For intra-cellular staining, cells were fixed and permeabilized according to the PerFix-nc kit (Beckman Coulter) manufacturer’s recommended protocol. After staining, cells were collected and analyzed using a Gallios Flow Cytometer (Beckman Coulter) equipped with three lasers (488 nm blue, 638 red, and 405 violet lasers) for the concurrent reading of up to 10 colors. The final data were analyzed using the Kaluza Flow Cytometry Analysis Software (Kaluza Analysis 2.1, Beckman Coulter).

To determine insulin-producing cells in mouse peripheral blood, we tested MIP-GFP transgenic mice according the approved animal protocol by institution IACUC committee. We purchased MIP-GFP transgenic mice from the Jackson Laboratory (Bar Harbor, ME, USA). The strain name is B6.Cg-Tg(Ins1-EGFP)1Hara/J (Stock Number: 006864).

### 2.4. Retinal Pigment Epithelium (RPE) Cell Differentiation of Mitochondrion-Induced PB-IPC (miPB-IPC)

To determine miPB-IPC’s multipotency and their RPE cell differentiation (Appendix A), miPB-IPC were treated with combined supplements (including L-glutamine, Gentamicin sulfate-Amphotericin (GA-1000), and basic fibroblast growth factor) in the presence of retinal pigment epithelial growth media (Lonza) for 8 days, in 24-well tissue culture-treated plates, at 37 °C in 5% CO_2_. The differentiated cells were characterized by immunocytochemistry with RPE-specific markers such as mouse anti-human mAbs RPE 65, CRALBP, and claudin-19, along with rabbit anti-tight junction protein 1 (ZO-1) polyclonal Ab (Novus Biological, Littleton, CO, USA). Human primary RPE cells were purchased from Lonza and served as positive control. Isotype-matched IgG served as negative control for immunostaining. For functional analysis, the phagocytosis of fluorescence latex beads (Sigma, Saint Louis, MO, USA) were performed in differentiated RPE cells. The phagocytosis-associated surface marker CD36 was examined by flow cytometry. The level of CD36 expression was quantified by mean fluorescence intensity after analyzed with Kaluza software version 2.1 (Beckman Coulter). 

### 2.5. Neuronal Differentiation of Mitochondrion-Induced PB-IPC (miPB-IPC)

To determine miPB-IPC’s multipotency and their neuronal cell differentiation (Appendix A), miPB-IPC were treated with 100 ng/mL neuronal growth factor (NGF, R & D Systems) + human neuronal stem cell growth medium (iXCells Biotechnologies, San Diego, CA, USA) for 3–5 days, in 24-well tissue culture-treated plates, at 37 °C in 5% CO_2_. The differentiated cells were characterized by immunocytochemistry [29] with mouse anti-human tyrosine hydroxylase monoclonal Ab (mAb, Clone LNC1, Catalogue # MAB 318, at 1:100 dilution) and rabbit anti-Synapsin I polyclonal Ab (Catalogue # AB1543, at 1:100 dilution) (EMD Millipore, Temecula, CA, USA). The FITC-conjugated AffiniPure donkey anti-mouse 2nd Ab and Cy3-conjugated AffiniPure donkey anti-rabbit 2nd Ab were purchased from Jackson ImmunoResearch Laboratories (West Grove, PA, USA). Isotype-matched IgG served as negative control for immunostaining. After covering with Mounting Medium with DAPI (Vector Laboratories, Burlingame, CA, USA), cells were photographed with Nikon A1R confocal microscope on Nikon Eclipse Ti2 inverted base. 

### 2.6. Colony Analysis

The miPB-IPC were initially cultured with the serum-free NutriStem^®^ hPSC XF culture medium (Corning) at 1 × 10^4^ cells/mL/well in 24-well tissue culture plates, at 37 °C in 8% CO_2_ culture condition. After miPB-IPC were cultured for 2 months, an individual colony was picked up under an inverted microscope by a BD Vacutainer Blood Collection Set (21G × ¾” × 12”) attached with a 3 mL syringe (Nipro, Miami, FL, USA) and inoculated in 96-well plates. Inspection by light microscope indicated ~80% of wells with a single colony. The wells with more than one colony were excluded [32]. The single colony (total *n* = 21 colonies) was manually dispersed by pipetting and aliquoted to 2–8 wells (depending on the size of colonies) for the induction of differentiation (Appendix A). The differentiation of single-colony-derived cells into different lineages were examined by using conditions respectively, as described above. For macrophage differentiation, three colony-derived cells were treated with M-CSF + HSC-Brew GMP Basal Medium for 2–3 days. For RPE cell differentiation, six colony-derived cells were treated with the RtEGM^TM^ Retinal Pigment Epithelial Cell Growth Medium BulletKit^TM^ (Lonza) for 3–5 days. For neuronal differentiation, the single colony-derived cells (total *n* = 9 colonies) were treated with 100 ng/mL neuronal growth factor (NGF, R & D Systems) + human neuronal stem cell growth medium (iXCells Biotechnologies, San Diego, CA, USA) in 96-well plates for 2–3 days. Their differentiations were evaluated with lineage-specific markers such as the phagocytosis of fluorescence latex beads for macrophage differentiation, RPE65 immunostaining for REP cells, and Synapsin I immunostaining for neuronal cells. Untreated colony-derived cells served as controls. Additionally, to determine their phenotype, the single-colony-derived cells (*n* = 3 colonies) were tested by flow cytometry with leukocyte common antigen CD45, HSC marker CD34, ES cell marker SOX2, together with memory cell marker CD45RO and CCR7. Isotype-matched IgGs served as control for flow cytometry.

To determine the multipotent differentiations of miPB-IPC, we initially performed colony analysis with three-germ layer-associated markers such as a neuronal marker synapsin for ectoderm, the islet β cell marker insulin for endoderm, and a macrophage marker CD11b for mesoderm. Additionally, using the 3-germ layer immunocytochemistry kit (Invitrogen, Carlsbad, CA, USA), we repeated the colony analysis by using Human Definitive Pancreatic Endoderm Analysis kit with additional three-germ layer-associated markers such as a neuronal marker beta III tubulin (Tuj1) for ectoderm, the liver cell marker alpha-fetoprotein (AFP) for endoderm, and smooth muscle actin (SMA) for mesoderm. For immunostaining, colonies were fixed and permeabilized in 24-well plates, and followed by immunostaining as described above. IgGs served as negative controls. 

### 2.7. Tumor Formation Assay

To determine the potential of tumor formation of miPB-IPC, the miPB-IPC were subcutaneously inoculated in the right flank of NSG mice (2 × 10^7^ cells per mouse in 200 µL physiological saline, S.C., right lower flank, *N* = 3 mice), according to the approved animal protocol at Hackensack Meridian Health. Injection of equal volume of physiological saline on the left lower flank served as control. Tumor formation and body weight were monitored once a week for 12 weeks. At the end of the observations, the liver, lung, spleen, and kidney tissues of miPB-IPC-treated mice were inspected and collected for histopathological examinations on tumor formation.

### 2.8. Tracking RFP-Labeled Mitochondria in PB-IPC

To directly examine the penetration of red fluorescent protein (RFP)-labeled mitochondria into PB-IPC, RFP-labeled mitochondria were purified from HEK-293 cell line after being labeled with CellLight™ Mitochondria-RFP BacMam 2.0 (Thermo Fisher Scientific, Waltham, MA, USA), according to manufacturer’s recommended protocol. PB-IPC were initially plated in 12 mm Nunc Glass Base Dish (Thermo Fisher Scientific) in NutriStem^®^ hPSC XF culture medium. After attaching for one hour, PB-IPC were treated with the purified RFP-labeled mitochondria in X-VIVO 15 medium (Lonza). After the treatment for 4 h, PB-IPC were photographed by using confocal microscopy. Hoechst 33,342 were applied to stain the nucleus of viable cells. 

### 2.9. Transmission Electron Microscopy (TEM)

To determine the penetration of mitochondria into nuclei, PB-IPC were treated with 100 μg/mL platelet-derived mitochondria for 12 h, at 37 °C in 8% CO_2_. Consequently, the mitochondrion-treated and untreated PB-IPC were collected at 500 g × 5 min and fixed with 2.5% glutaraldehyde/4% paraformaldehyde in 0.1 M Cacodylate buffer for transmission electron microscope (Philips CM12 electron microscope with AMT-XR11 digital camera) analysis. Alternatively, the purified viable nuclei were labeled with Hoechst 33,342 and were incubated with MitoTracker Deep Red-labeled mitochondria. Their interactions were directly observed and photographed under confocal microscope. 

### 2.10. Blocking Experiment with CXCR4 Receptor Antagonist AMD 3100

To determine whether the action of SDF-1/CXCR4 contributed to the penetration of mitochondria into nuclei, we performed the blocking experiment with CXCR4 receptor antagonist AMD3100. The purified PB-IPC’s nuclei were treated with MitoTracker Deep Red-labeled purified mitochondria in the presence or absence of AMD 3100 (30 μM). The equal concentration of solvent DMSO served as control. After the treatment for 4 h, nuclei were washed twice with PBS and prepared for flow cytometry.

### 2.11. Quantitative Real Time PCR

To clarify the interaction of mitochondria and nuclei, PB-IPCs’ nuclei were isolated using Nuclei isolation kit (Sigma) according to the manufacturer’s recommended protocol. PB-IPCs’ nuclei were treated with 100 μg/mL platelet-derived mitochondria for 4 hrs, at 37 °C in 8% CO_2_. Consequently, the mitochondrion-treated and untreated nuclei were collected at 500 g × 5 min and fixed with 2.5% glutaraldehyde/4% paraformaldehyde in 0.1 M cacodylate buffer for electronic microscope. Additionally, RT^2^ Profiler real time PCR Array was applied to study the directly genetic and epigenetic modulations of mitochondria by using the Human Epigenetic Chromatin Modification Enzymes kit (96-well format, Qiagen, Valencia, CA, USA). RT^2^ Profiler real time PCR Array was used according to the manufacturer’s instructions. The data were analyzed using PrimePCR array analysis software (Bio-Rad, Hercules, CA, USA). 

To detect the expression of human islet β-related gene markers by quantitative real time PCR, PB-IPC were isolated from culture vessels after attachment at different time points such as 6, 12, 24, 48, and 72 hrs. Total RNAs from each sample were extracted using a Qiagen kit (Valencia, CA, USA). First-strand cDNAs were synthesized from total RNA using an iScript gDNA Clear cDNA Synthesis Kit according to the manufacturer’s instructions (Bio-Rad, Hercules, CA, USA). Real-time PCR was performed on each sample in triplicate using the StepOnePlus Real-Time PCR System (Applied Biosystems, CA, USA) under the following conditions: 95 °C for 10 min, then 40 cycles of 95 °C for 15 s, and 60 °C for 60 s, using the validated gene-specific PCR Primer sets for each gene including pancreatic islet cell-related markers including insulin (Bio-Rad Laboratories, Hercules, CA, USA), MAFA, NKX6.1, and PDX-1 (Qiagen, Valencia, CA, USA). The expression level of each gene was determined relative to β-actin as an internal control. To confirm gene expression, real time PCR products were examined with 1.5% agarose gel electrophoresis. 

### 2.12. RNA-seq

RNA sequencing (RNA-seq) analysis was performed between the mitochondrion-treated and untreated PB-IPC in four preparations. Total RNAs from each sample were extracted using a Qiagen kit (Valencia, CA, USA) and shipped to Genewiz (South Plainfield, NJ, USA) in dry ice for standard RNA sequencing and profiling gene expression by using Illumina NovaSeq™ 6000 Sequencing System (Genewiz, South Plainfield, NJ, USA), with 2 × 150 bp configuration, single index, per lane.

### 2.13. Statistics

Statistical analyses of data were performed by the two-tailed paired Student’s *t*-test to determine statistical significance between untreated and treated groups. Values were given as mean ± SD (standard deviation).

## 3. Results

### 3.1. Adult Peripheral Blood-Derived PB-IPC Display Human Islet β Cell-Specific Markers 

To characterize the specific marker of PB-IPC, they were purified from adult peripheral blood by virtue of their ability to attach to the hydrophobic plastic surface of Petri dishes in serum-free culture medium. Flow cytometry demonstrated that PB-IPC displayed the phenotype of Lin^-^CD34^-^CD45^+^SOX2^+^CD45RO^+^CCR7^+^ (Figure 1A), including the expression of leukocyte common antigen CD45, memory cell markers CD45RO and CCR7, along with ES cell marker SOX2 and low CD117 expression. In contrast, they were negative for HSC markers CD34 and CD38, T cell markers (e.g., CD3, CD4, and CD8), B cell marker CD19, granulocyte marker CD66b, and MK/platelet markers CD41 and CD42a (Figure 1B). CD14^+^ monocytes/macrophages which could not adhere to the hydrophobic surface of culture vessels underwent apoptosis and/or necrosis within 24-h of culture (Figure 1C). Additionally, we analyzed cell cycles of the freshly-isolated PB-IPC after overnight attachment by flow cytometry with propidium iodide (PI) staining. The data demonstrated that there was 0.9 ± 0.5% of freshly-isolated PB-IPC distributed in the S phase, with 92.94 ± 2.75% in G_0_/G_1_ phases and 6.68 ± 2.2% in G_2_/M phases (Figure 1D). This indicates the limited potential of cellular proliferation for the freshly-isolated PB-IPC. Thus, PB-IPC display a unique phenotype and are different from mesenchymal stem cells (MSC) [34] and our previously characterized monocyte-derived stem cells (designated fibroblast-like macrophages, f-Mφ) [32].

Next, we determined the insulin production of PB-IPC. Using human islet cells as a positive control group, real time PCR data revealed that PB-IPC expressed human islet β cell-specific markers such as insulin and transcription factors (PDX-1, NKX6.1, and MAFA) mRNAs (Figure 1E). Kinetic analysis noticed that these gene markers were stable in most PB-IPC samples within 24-h ex vivo cultures of PB-IPC in the presence of serum-free X-VIVO 15 media, while some markers were disappeared or down-regulated, due to the potential different health statuses of blood donors. Flow cytometry further confirmed the double-positive cells with expressions of MAFA and C-peptide (a by-product of insulin) at protein levels (Figure 1F). Since MAFA is the only islet β cell-specific activator responsible for insulin expression [35], and glucose transporter 2 (GLUT2) is a surface marker for human islet β cells, we further analyzed the percentage of PB-IPC in human peripheral blood mononuclear cells (PBMC) by using MAFA + GLUT2 in combination with the above PB-IPC’s markers, with an additional marker for an ES cell-associated transcription factor octamer-binding protein 3/4 (OCT3/4) and SRY-box containing gene 2 (SOX2). Flow cytometry analysis indicated that there was 0.0045 ± 0.004 of Lin1^-^CD34^-^CD45^+^CD45RO^+^CCR7^+^SOX2^+^OCT3/4^+^ MAFA^+^Glut2^+^ PB-IPC cells in freshly Ficoll Paque-isolated human PBMC. After overnight (12 hrs) attachment selection, PB-IPC can be isolated from PBMC and display the same phenotype, with expression of Lin1^-^CD34^-^CD45^+^CD45RO^+^CCR7^+^SOX2^+^OCT3/4^+^ MAFA^+^Glut2^+^ (Figure 1G). Additionally, we also found GFP-positive insulin-producing cells in the peripheral blood of insulin promotor-green fluorescence protein (GFP)-transgenic mice (The strain name: B6.Cg-Tg(Ins1-EGFP)1Hara/J, stock No: 006864) (Figure 1H). Therefore, these data established the existence of PB-IPC in peripheral blood that can be isolated by the current approach. 

### 3.2. Ex Vivo Differentiation of Mitochondrion-Induced PB-IPC (miPB-IPC) into Retinal Pigment Epithelium (RPE) Cells 

Platelets are enucleate cells without human genomic DNA. We obtained the apheresis platelets from New York Blood Center, with high purity (>99% of CD41^+^CD42^+^ platelets [29]) for our studies. The purity of isolated mitochondria was ≥90%. A previous study demonstrated that treatment with platelet-derived mitochondria can improve the proliferation and function of human pancreatic islet β cells [29]. To improve the insulin production and β-cell differentiation of PB-IPC, we prepared the purified platelet-derived mitochondria from autologous or allogeneic peripheral blood, as described previously [29] and used this to treat PB-IPC isolated and expanded from blood samples of adult donors at the New York Blood Center (*n* = 42; mean age of 47.64 ± 14.07; age range from 16 to 73 years old; 23 males and 19 females). Unexpectedly, we found that PB-IPC could gave rise to other cell lineages in the presence of different inducers. 

The RPE is a monolayer cell that fundamentally supports visual function and the integrity of photoreceptors. Dysfunctions and loss of RPE cells is the major cause for age-related macular degeneration (AMD), leading to blindness [36,37]. To determine whether miPB-IPC were multipotent, we tested their differentiation to RPE cells. Treatment with combined supplements (including L-glutamine, Gentamicin sulfate-Amphotericin (GA-1000), and basic fibroblast growth factor) in the presence of RPE growth media for 8 days caused >90% of miPB-IPC to acquire the RPE phenotype, such as pigmented granules in the cytoplasm, numerous cell processes at various lengths (Figure 2A), and expression of visual cycle proteins RPE65 and cellular retinaldehyde binding protein (CRALBP), as well as the tight junction-associated membrane proteins claudin-19 and Zonular occludens-1 (ZO-1) (Figure 2B, bottom), similar to primary human RPE cells (Figure 2B, top). Functional analysis yielded a strong phagocytosis of fluorescence beads (Figure 2C) and an up-regulated expression of phagocytic marker CD36 (Figure 2D), similar to human RPE cells [38]. While untreated cells failed to show these changes. These results indicated the differentiated RPE cells acquired the phenotype of human RPE cells. 

### 3.3. Ex Vivo Differentiation of Mitochondrion-Induced PB-IPC (miPB-IPC) into Neuronal Cells 

During the induction of RPE cell differentiation, we observed a few elongated neuronal-like cells and therefore tested the differentiation potential of miPB-IPC to neuronal cells. After treatment with 100 ng/mL neuronal growth factor (NGF) + human neuronal stem cell growth medium in 24-well plates for 2–3 days, 99% of treated miPB-IPC displayed typical neuronal morphology including elongated axon-like processes with branches and formed cell–cell networks via dendrites (Figure 3A). Double-immunostaining revealed that 99.1% of treated cells expressed the neuronal-specific marker synapsin I and tyrosine hydroxylase (Figure 3B), a rate-limiting enzyme for the biosynthesis of catecholamines (e.g., dopamine and norepinephrine) [39]. Untreated cells only showed a spontaneous differentiation (<3%). These data indicate the adrenergic-neuronal differentiation potential of miPB-IPC. 

### 3.4. Clonal Analysis of miPB-IPC

To further determine the multipotency of miPB-IPC, we performed clone analysis [32]. We observed colony formation of miPB-IPC with different sizes (Figure 4A) and potential for colony formation of miPB-IPC was markedly increased after treatment with mitochondria (Figure 4B). Flow cytometry verified these colonies retained PB-IPC markers such as CD45^+^ and CD34^-^ (94.7 ± 4.29%, *n* = 3), SOX2^+^ (77.38 ± 13.34%), together with CD45RO^+^ and CCR7^+^ (92.4 ± 3.6%) (Figure 4C). We dispersed five colonies and inoculated them into 96-well plates. After treatment with different lineage-specific inducers including 50 ng/mL M-CSF for macrophage differentiation, 100 ng/mL NGF for neuronal cells, and RPE cells with specific condition medium, characterization with different lineage markers substantiated that 62.05 ± 6.43% of differentiated Mφ exhibited phagocytosis of fluorescence beads (Figure 4D, left), 75.6 ± 4.8% of RPE65-positive cells (Figure 4D, middle), 94.8 ± 1.7% of synapsin I-positive neuronal cells, with distinctive morphologies (Figure 4D, right). Untreated cells showed minimal spontaneous differentiation (<5%). These data demonstrate that single-colony-derived cells can give rise to different cell lineages such as macrophages, RPE cells and neuronal cells, confirming the multipotent nature of miPB-IPC. 

Additional study confirmed no tumor formation after transplant of miPB-IPC at the dose of 2 × 10^7^ cells/mouse (s.c.). The miPB-IPC-transplanted mice gained weight at the time of a 12-week follow-up (Figure 4E, *n* = 3 mice), without evident tumor formation upon tissue inspection (lung, liver, spleen, and kidney), indicating the safety of the miPB-IPC application.

To determine the multipotent differentiations of miPB-IPC, we performed colony analysis with three-germ layer-associated markers including a neuronal marker synapsin for ectoderm, the islet β cell marker insulin for endoderm, and a macrophage marker CD11b for mesoderm. Confocal microscopy demonstrated that there were more three-germ layer-positive cells distributed in the miPB-IPC-derived colonies (Figure 4F) than those in the mitochondrion-untreated PB-IPC-derived colonies (Figure 4F). Using the three-germ layer immunocytochemistry kit (Invitrogen), we repeated colony analysis with additional three-germ layer-associated markers such as a neuronal marker beta III tubulin (Tuj1) for ectoderm, the liver cell marker alpha-fetoprotein (AFP) for endoderm, and smooth muscle actin (SMA) for mesoderm. The data confirmed the spontaneously differentiated three-germ layer-positive cells in the miPB-IPC-derived colonies (Figure 4G). The number of positive cells was very low or negative in the mitochondrion-untreated PB-IPC-derived colonies (Figure 4G). Thus, the data proved the multipotency of miPB-IPC. 

### 3.5. Penetration of Mitochondria into Nuclei of PB-IPC

To explore the action of exogenous platelet-derived mitochondria in PB-IPC, through electron microscopy, we observed that mitochondria migrate to the nuclei of PB-IPC (Figure 5A,B) following the treatment conditions. We observed mitochondria crossing the nuclear membrane (Figure 5A), located inside nuclear matrixes and close to the nucleolus (Figure 5B), as well as with a similar shape of mitochondrion in the cytoplasm (Figure 5B, indicated by red arrow). Untreated PB-IPC failed to show such marked phenomena (Figure 5C). To further confirm the penetration of exogenous mitochondria into PB-IPC’s nuclei, PB-IPC were treated with red fluorescent protein (RFP)-labeled mitochondria (Figure 5D), which were isolated from HEK 293 cell line. After the treatment for 4 h, confocal microscopy established RFP^+^ mitochondria infiltrating the cytoplasm (Figure 5D). To directly visualize the interaction between the mitochondria and the nucleus, freshly-purified PB-IPC-derived nuclei were treated with isolated MitoTracker Red-labeled mitochondria. Confocal imaging revealed the direct interaction of mitochondria with nuclei, while some labeled mitochondria entered nuclei (Figure 5E). Based on the observation under transmission electronic microscope (TEM) and flow cytometry by staining with the mitochondrial markers such as MitoTracker Deep Red staining, anti-cytochrome C and anti-heat shock protein (HSP) 60 mAbs, the frequency of intra-nuclear mitochondria was about 1–3%.

Next, we explored the molecular mechanisms underlying the migration of mitochondria to nuclei. Flow cytometry demonstrated that nuclei displayed the chemokine receptor CXCR4 (Figure 5F, the ligand for stromal cell-derived factor (SDF)-1), while mitochondria expressed SDF-1 (Figure 5G). To determine whether the action of SDF-1/CXCR4 contributed to the penetration of mitochondria into nuclei, we performed the blocking experiment with CXCR4 receptor antagonist AMD3100. The purified PB-IPC’s nuclei were treated with MitoTracker Deep Red-labeled purified mitochondria in the presence or absence of AMD 3100. After the treatment for 4 hrs, flow cytometry demonstrated that the percentage of MitoTracker Deep Red-positive nuclei was markedly reduced after the treatment with AMD 3100 (Figure 5H). It indicated that mitochondria entered into nuclei through the chemoattractant interactions between SDF-1 and CXCR4.

### 3.6. Genetic and Epigenetic Changes in PB-IPC after the Treatment with Mitochondria

To investigate the mechanism by which mitochondria modulate expression in the nucleus, we treated purified PB-IPC-derived viable nuclei with isolated mitochondria for 4 h at 37 °C and 5% CO_2_ and assessed changes in transcription by real-time PCR array. The data revealed marked changes in epigenetic chromatin modification enzyme-related genes such as histone acetyltransferase lysine acetyltransferase 2B (*KAT2B*), histone methyltransferase mixed lineage leukemia (*MLL*), histone phosphorylation-related p21 protein-activated kinase 1 (*PAK1*), and histone deacetylases (histone deacetylase 3 (*HDAC3*), *HDAC4*, *HDAC5*, and *HDAC6*) (Appendix A). These data demonstrate that mitochondria that penetrate the nucleus contribute to both epigenetic and genetic regulations, leading to the reprogramming of PB-IPC. To find more differentially expressed genes, we performed RNA sequencing (RNA-seq) analysis between the mitochondrion-treated and untreated PB-IPC in four preparations (Figure 6A). The results demonstrated that 37 genes were markedly up-regulated in the mitochondrion-treated PB-IPC (Figure 6B, *p* < 0.05), and 9 genes were down-regulated (Figure 6C, *p* < 0.05). There were no significant changes for other genes (*N* = 15,388, 99.7% of genes) in PB-IPC after the treatment with mitochondria.

## 4. Discussion

Adult stem cells are a rare population found in all tissues to replace cells or tissues that are damaged by diseases or injury. The current study demonstrated the existence of PB-IPC in human peripheral blood, displaying a unique phenotype (Lin1^-^CD34^-^CD45^+^CD45RO^+^CCR7^+^ SOX2^+^OCT3/4^+^ MAFA^+^Glut2^+^). Using our current established protocol, PB-IPC were grown and expanded by adhering to the hydrophobic bottom of Petri dishes in chemical-defined serum-free culture without adding any other growth factors. Notably, the differentiation potential of PB-IPC was markedly increased after the treatment with platelet-derived mitochondria, giving rise to three-germ layer-derived cells. The miPB-IPC exhibited high efficiency of differentiations toward RPE and neuronal cells in the presence of different inducers, respectively, confirming the multipotency of PB-IPC post-mitochondrial treatment. Thus, these cells offer great promise as a solution for the current bottlenecks associated with conventional stem cell transplants and have tremendous potential for patient benefit in the clinic. 

To circumvent the limitation of adult stem cells, functional tissue cells (e.g., insulin-producing cells or islet cells) have been generated from embryonic stem (ES) cells and induced pluripotent stem cells (iPS) through ex vivo induction of differentiations [20,40,41,42]. However, recent advance in stem cell biology have realized that ES cells, iPS, and their derived cells can also cause immune rejections post transplantation [43,44,45], challenging their clinical therapeutic potentials [19]. To avoid immune cells attack on transplanted cells and/or cell-delivery devices, and to provide sufficiently-permeabilized nutrients to sustain their cell viability, an encapsulation with different biomaterials or the use of a semipermeable membrane/capsule have been evaluated through animal and pilot clinical studies. However, formation of fibrosis (scarring around the device) causing a death of capsulized cells and failure of encapsulation devices is still a big roadblock, and the optimal device remains to be developed [46]. Current and previous studies demonstrated PB-IPC naturally circulate in human peripheral blood, displaying islet β cell-related markers and reducing hyperglycemia with migration to pancreatic islets after transplant into the chemical streptozotocin (STZ)-induced diabetic mice [30]. Our findings provide a novel approach for the generation of a large amount of autologous insulin-producing cells from patients themselves to potentially treat diabetes in clinics after optimizing ex vivo culture conditions. In comparison with the generation of insulin-producing cells from ES and iPS cells, this technology can efficiently isolate insulin-producing cells from their own blood, without any ethical issues and the hazards of immune rejection. Moreover, multipotent differentiation of miPB-IPC into other cell lineages will provide unmet medical needs to circumvent those limitations not only for diabetic subjects, but also for the whole field of regenerative medicine.

Mitochondria are normally considered as organelles in the cytoplasm, but the existence of intra-nuclear mitochondrion (inMito) is an unrecognized physiological fact. Intra-nuclear mitochondria have been reported in leukemic cells [47], fibroblasts of normal cow liver capsules [48], and other diseased tissues [49,50]. We found the penetration of mitochondria into nuclei, as demonstrated by transmission electron and confocal microscopes. Using purified viable nuclei of PB-IPC, real-time PCR array further established the significant contribution of mitochondria to epigenetic and genetic modulations of nuclei within only 4 hr, highlighting the directly-rapid interaction between mitochondria and nuclei and leading to the cell reprogramming. Additionally, we examined that PBMC-derived mitochondria (not platelets) could also penetrate the nucleus of PB-IPC with the incidence of 2.09 ± 0.87%. Flow cytometry revealed that PBMC-derived mitochondria displayed the similar level of SDF-1 expression as that of platelet-derived mitochondria, but much higher than that of PB-IPC-derived mitochondria (Appendix A). Taken together with the results of blocking with CXCR4 receptor antagonist AMD 3100 on the purified nuclei of PB-IPC, these data indicate that SDF-1/CXCR4 pathway contributes to the migration of mitochondria to the nuclear membrane of PB-IPC, leading to the penetration of the nucleus of PB-IPC. Increasing evidence demonstrated that mitochondria play a key role during the reprogramming of somatic cells to iPS cells, as well as the self-renewal and pluripotent differentiations of ES cells [51]. The detailed molecular mechanisms should be further determined to better understand the reprogramming of PB-IPC. In conclusion, this study is the first to report multipotent stem cells can be produced from adult human peripheral blood after the treatment with platelet-derived mitochondria, advancing the stem cell technology for regeneration medicine.

## Figures and Tables

**Figure 1 cells-09-01350-f001:**
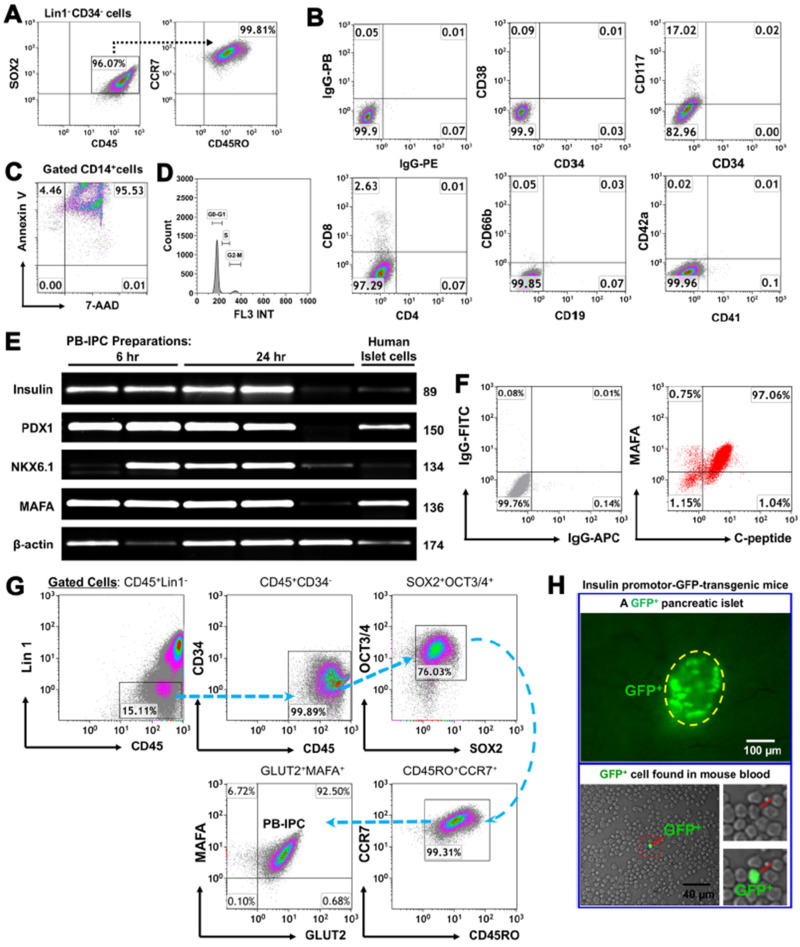
Characterization of peripheral blood-derived insulin-producing cells (PB-IPC) from adult peripheral blood with islet β-cell-related markers. (**A**) Characterization of PB-IPC. Peripheral blood-derived mononuclear cells (PBMC) were plated in Petri dishes in the presence of serum-free culture medium. After attachment overnight (12 hrs), PB-IPC were isolated by removing all floating cells and debris. The gated Lin1^-^CD34^-^ cells express CD45, SOX2, CD45RO, and CCR7. Representative data from eight preparations. (**B**) Phenotype of PB-IPC, with low expression of CD117, but no expression of CD4, CD8, CD19, CD34, CD38, CD41, CD42a, and CD66b. Isotype-matched IgGs served as controls (*n* = 8). (**C**) Apoptosis (Annexin V^+^) and necrosis (7-AAD^+^) of blood monocytes after 24-h culture in non-tissue culture-treated Petri dishes (*n* = 5). (**D**) Analysis of cell cycles in the freshly-isolated PB-IPC after overnight attachment by flow cytometry with propidium iodide (PI) staining (*n* = 5). (**E**) Real time PCR analysis of pancreatic islet β-cell-related markers in PB-IPC isolated from healthy donors (*n* = 5). Freshly isolated human islets served as positive controls. (**F**) Flow cytometry for islet β-cell-related transcription factor MAFA and an insulin by-product C-peptide by double immunostaining (*n* = 5). (**G**) Flow cytometry for determining PB-IPC’s phenotype after overnight attachment. Representative data from four preparations. FITC-conjugated anti-human lineage cocktail 1 (Lin1) (CD3, CD14, CD16, CD19, CD20, CD56) was applied to eliminate the known cell lineages such as T cells, monocytes/macrophages, granulocytes, B cells, and natural killer (NK) cells. Anti-human leukocyte common antigen CD45 mAb was used to remove the red blood cells (RBC) and platelets’ contamination during data analysis. MAFA (transcription factor) and GLUT2 (β cell surface marker) were utilized to determine the islet β cell-associated phenotype in PB-IPC. (**H**) Fluorescence microscopy shows a GFP^+^ cell among PBMC of an insulin promotor 1-GFP-transgenic mouse (*n* = 3). A GFP-positive mouse islet served as positive control.

**Figure 2 cells-09-01350-f002:**
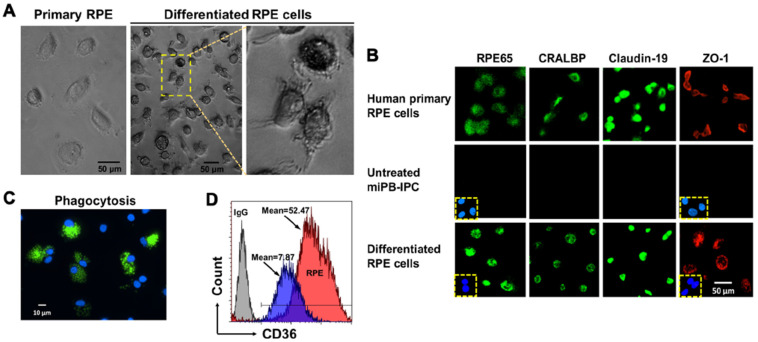
Differentiation of mitochondrion-induced PB-IPC (miPB-IPC) into retinal pigment epithelium (RPE) cells. (**A**) Phase contrast images show the differentiation of miPB-IPC into RPE cells with cellular pigmentation and processes at varied lengths (*n* = 5). (**B**) Immunostaining of differentiated RPE cells with RPE-specific markers (*n* = 3). The human primary RPE cells served as positive controls. Mouse IgG and rabbit IgG merged with nuclear DAPI (blue) staining served as negative controls (inserts). Untreated miPB-IPC served as negative control (middle panel). (**C**) Phagocytosis of fluorescence beads by differentiated RPE cells (*n* = 3). (**D**) Flow cytometry analysis of CD36 expression on differentiated RPE cells (red) and untreated cells (blue) (*n* = 3). Isotype-matched IgG served as negative control (grey).

**Figure 3 cells-09-01350-f003:**
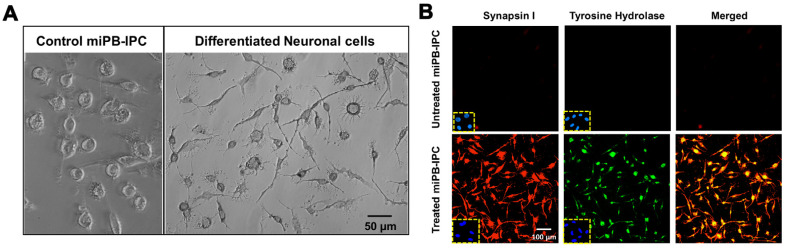
Differentiation of miPB-IPC into neuronal cells. (**A**) Phase contrast images show the differentiation of miPB-IPC into neuronal cells (*n* = 5). (**B**) Immunostaining of differentiated neuronal cells with neuron-specific markers synapsin I (red) and tyrosine hydroxylase (green) (*n* = 3). IgG staining served as negative control (inserts). Untreated miPB-IPC served as negative control (top panel).

**Figure 4 cells-09-01350-f004:**
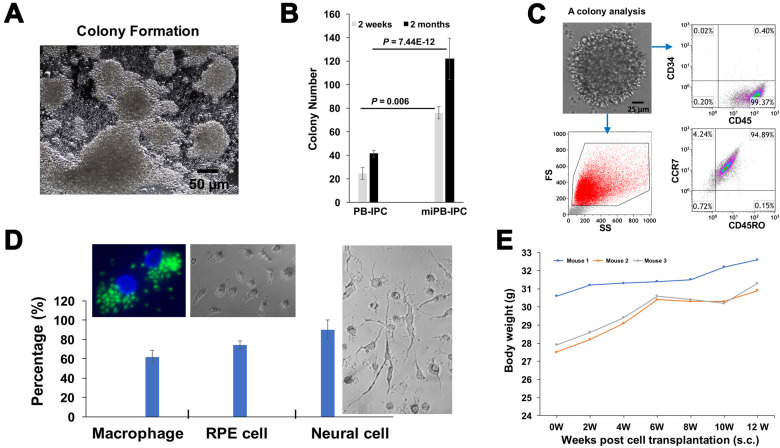
Clonal analysis of miPB-IPC and testing for tumor formation of miPB-IPC. (**A**) Colony formation of miPB-IPC with different sizes in regular miPB-IPC cell cultures (*n* = 5). (**B**) Potential difference in colony formation between miPB-IPC relative to untreated PB-IPC. Increase the colony formation in miPB-IPC relative to untreated PB-IPC. The colony formation of miPB-IPC with different sizes at 2-month culture in 24-well plate. The miPB-IPC were initially cultured with the serum-free NutriStem^®^ hPSC XF culture medium (Corning) at 1 × 10^4^ cells/mL/well in 24-well tissue culture plates, at 37 °C in 8% CO_2_ culture condition. Data are presented as mean ± SD from five preparations. (**C**) Phenotypic analysis of single colony-derived cells, retaining the PB-IPC’s markers CD34^-^CD45^+^SOX2^+^CD45RO^+^CCR7^+^ (*n* = 5). (**D**) Clonal analysis. A single colony was dispersed and inoculated into 96-well plates, treated wells with different lineage-specific inducers for differentiations, including macrophages (left, phagocytosis of green fluorescence beads, *n* = 3), RPE cells (middle, *n* = 6), and neural cells (right, *n* = 9). (**E**) Gaining weight in miPB-IPC-transplanted mice, without tumor formation. The miPB-IPC at the dose of 2 × 10^7^ cells/mouse were inoculated (s.c., right lower flank) in NOD-scid IL-2Rγ^null^ mice (*n* = 3). Injection of equal volume of physiological saline on the left lower flank served as control. (**F**) Colony analysis with three-germ layer-associated markers such as a neuronal marker synapsin I for ectoderm, the islet β cell marker insulin for endoderm, and a macrophage marker CD11b for mesoderm. IgGs served as negative controls (top panel). Representative images were from one of eight colonies for miPB-IPC group (bottom panel) and five colonies for control PB-IPC (middle panel). (**G**) Colony analysis with additional three-germ layer-associated markers such as a neuronal marker beta III tubulin (Tuj1) for ectoderm, the liver cell marker alpha-fetoprotein (AFP) for endoderm, and smooth muscle actin (SMA) for mesoderm. IgGs served as negative controls (top panel). Representative images were from one of seven colonies for miPB-IPC group (bottom panel) and five colonies for control PB-IPC (middle panel).

**Figure 5 cells-09-01350-f005:**
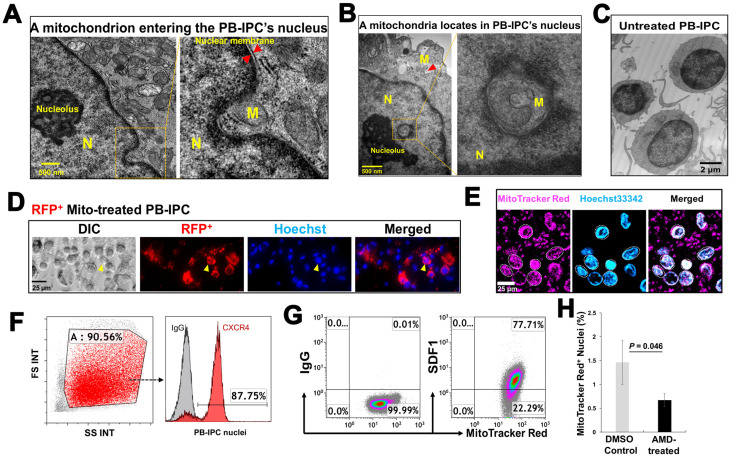
Penetration of mitochondria into nuclei of PB-IPC. (**A**) Transmission electron microscopy demonstrates a mitochondrion (M) crossing the nuclear membrane of a mitochondrion-treated PB-IPC. (**B**) A mitochondrion located inside the nuclear matrix, and close to the nucleolus with a morphologically-similar mitochondrion (indicated by red arrow) in the cytoplasm. (**C**) Ultrastructure of untreated PB-IPC. (**D**) Penetration of red fluorescent protein (RFP)-labeled mitochondria into PB-IPC. After PB-IPC were treated with RFP-labeled mitochondria for 4 h, confocal microscopy established RFP^+^ mitochondria infiltrating the cytoplasm (*n* = 5). Distribution of RFP^+^ mitochondria inside of a nuclear was represented by an orange arrow. RFP^+^ mitochondria (red) were colocalized with the Hoechst 33342-labeled nuclear (blue) and the differential interference contrast (DIC) image (left). (**E**) MitoTracker Red-labeled mitochondria (pink) entered nuclei (blue), and their colocalization was shown by confocal microscopy (*n* = 5). Isolated mitochondria from platelets were co-cultured with purified nuclei of PB-IPC for 4 h in the presence of serum-free culture medium X-VIVO 15 at 37 °C and 5% CO_2_. (**F**) Expression of CXCR4 on the membrane of purified nuclei (*n* = 4). (**G**) Mitochondria displaying CXCR4 ligand SDF-1 (*n* = 4). (**H**) Blocking experiment with CXCR4 receptor antagonist AMD 3100. The purified PB-IPC’s nuclei were treated with MitoTracker Red-labeled purified mitochondria in the presence or absence of AMD 3100 (30 μM, *n* = 3). The equal concentration of solvent DMSO served as control. After the treatment for 4 hrs, nuclei were washed twice with PBS and prepared for flow cytometry.

**Figure 6 cells-09-01350-f006:**
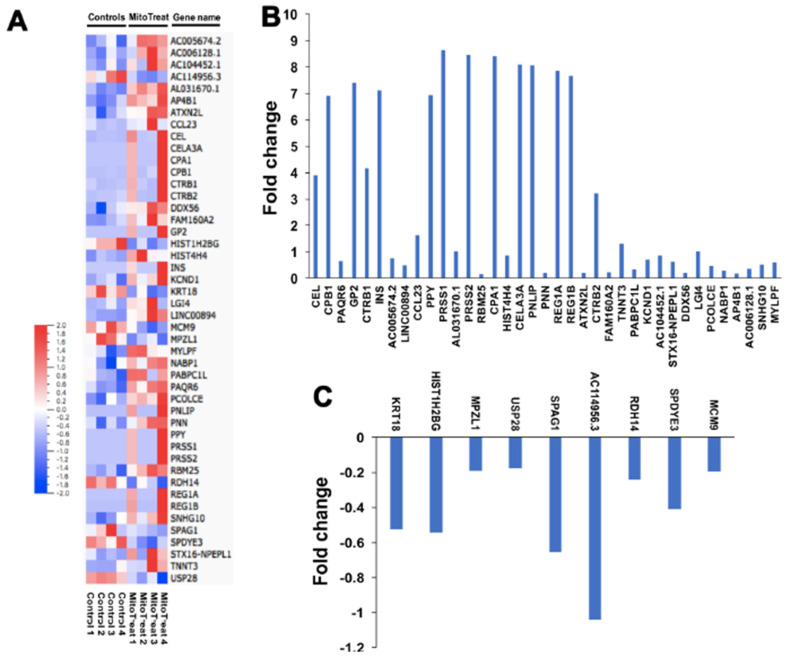
RNA-seq analysis in PB-IPC. Differentially expressed genes shown by the RNA-seq in four PB-IPC preparations treated with mitochondria. Untreated PB-IPC served as controls. (**A**) The heatmap revealed that forty-six most differentially expressed genes in PB-IPC post the treatment with mitochondria. (**B**) Thirty-seven up-regulated genes in PB-IPC post the treatment with mitochondria. (**C**) Nine down-regulated genes in PB-IPC post the treatment with mitochondria.

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
