# Peer review of "Generation of Multipotent Stem Cells from Adult Human Peripheral Blood Following the Treatment with Platelet-Derived Mitochondria"

_cells, 2020, doi:10.3390/cells9061350_

Round 1
Reviewer 1 Report
The study is very interesting and the article is well written.
I suggest expanding the backgroung and discussion of the importance of stem cells in the therapeutic approach. For example, using the review on the topic: Int J Mol Sci. 2019 May 7;20(9).
Author Response
Dear Reviewer,
We appreciate your kind consideration! We already revised both sections you have suggested.
Best regards,
Yong
Reviewer 2 Report
Review of “Generation of multipotent stem cells from adult human peripheral blood following the treatment with platelet-derived mitochondria” by Yu et al.
Summary:
The authors investigate the pluripotency of adult peripheral blood-derived insulin-producing cells (PB-IBC) that were treated with mitochondria isolated from blood platelets. This study is relevant to determining the clinical applicability of using an individual’s circulating blood stem cells to replenish stem cells elsewhere in the body following ex vivo “education”. The authors found that exposure to both isolated platelet mitochondria as well as necessary growth factors will induce the PB-IBC cells to differentiate into cell types from 3 distinct germ layers (including macrophage, retinal pigment epithelial cells, and neuronal cells). Additionally, injection of these cells into immunocompromised mice did not lead to tumor formation after 12 weeks. Finally, the authors explored the possible mechanism behind mitochondrial exposure leading to multipotency and observed that a fraction of the mitochondria would penetrate the nucleus of the PB-IBCs, which was then shown to contribute to changes in gene expression. The findings of mitochondria-induced multipotency are novel and provide promising support for further exploration of this method for clinical purposes.
Major criticisms/questions:
The methodology used to generate the multipotent cells from PB-IBCs is unclear. The authors do not clarify whether the treatment with mitochondria is separate from the addition of other inducing factors for differentiation (i.e. treatment begins and ends prior to additional factors), or whether these are provided to the cells at the same time.
Additionally, the mechanism for the induced multipotency has been suggested to be the penetration of mitochondria into the nucleus of PB-IBCs. However, the authors only reported seeing this occur in 1-3% of the cells. Is this rate of occurrence enough to be biologically meaningful? And with this incidence already being so low, is the significance of its “inhibition” by AMD3100 meaningful? It would be interesting to see whether the inhibitor AMD3100 would significantly disrupt the changes to gene expression that were shown by the authors to occur in PB-IBCs treated with mitochondria.
Minor Revisions:
- In the introduction, it would be helpful to elaborate on the nature of PB-IBCs (when they were first discovered, what their proposed function is, etc.)
- Although in the methods PCR is reported to be performed at several time points (6,12,24,48, and 72 hours), data from only 2 time points are shown in Figure 1. Is there some justification for this?
- The authors do not specify how the mitochondrial purity was determined to be 90%.
- Figure 3B would be strengthened by a human control neuronal cell, as the human equivalent was provided for retinal epithelium differentiation.
- There are several spelling errors or unclear wording throughout the manuscript, including:
Line 20: “cell” should be changed to “cells”
Line 57: When it says “platelet-releasing mitochondria”, is this supposed to mean mitochondria released by platelets? The current wording makes this point unclear.
Line 60: Please provide a definition for adult peripheral blood insulin-producing cells.
Line 62: Have been treated by other groups or were treated in the present study/by this group in another paper? Unclear wording.
Line 105: “nuclear” should be “nuclei”
Line 116: “anti-huuman” should be “anti-human”
Line 133: Define “RPE” here
Line 170: “followed by testing of phagocytosis” is redundant given the explanation of testing that comes in line 175.
Line 297: “maker” should be “marker”
Line 302: What is the unit for the “0.0045” value?
Figure 4E: Each line should be labeled “mouse1”, “mouse 2”, etc. instead of “mice 1”, “mice 2”
Author Response
Dear Reviewer,
We appreciate your kind consideration! Please see the attached document.
Best regards,
Yong

Reviewer 3 Report
The manuscript of Yu H. et al. is devoted to the generation of multipotent stem cells from adult peripheral blood-derived insulin-producing cells (PB-IPC) after the treatment with platelet-derived mitochondria. The study demonstrated that mitochondrion-induced PB-IPC can differentiate into a number of specialized cells. It was shown that platelet-derived mitochondria can penetrate the nucleus of PB-IPC and, thus, cause epigenetic changes. The study is very interesting; the results can be useful for clinical applications.
Major comments
- A significant part of the Introduction section is devoted to type 1 and type 2 diabetes mellitus. At the same time, the work examines the multipotent capability of the differentiation of mitochondrion-induced PB-IPC into different types of cells. The authors should revise the Introduction section, paying more attention to the problems associated with the generation of multipotent stem cells. It should be explained why exactly PB-IPC were chosen to form multipotent stem cells. In previous studies, the authors found that PB-IPC without any treatment can differentiate into multipotent stem cells. What is the reason for the treatment with platelet-derived mitochondria? All these issues should be described in the Introduction.
- The authors present data that platelet-derived mitochondria penetrate the nucleus of PB-IPC, which can lead to the cell reprogramming. It is necessary to discuss whether platelet-derived mitochondria have features compared to PB-IPC mitochondria. The difference between platelet-derived mitochondria, which have such a penetrating ability, from the original mitochondria of PB-IPC should be clarified. Please, provide an explanation of what this ability of mitochondria derived from platelets may be related to. Can mitochondria of other types of cells (not platelets) cause similar processes?
- The figures should be described in more detail. For example, why is there no expression level of some genes in Fig. 1E? Which cell culture of the five ones described above was used in further experiments? What process is indicated by the dashed arrow in Fig. 1G? All this will help better understand the data described in the text.
Minor comments
- There are some typographic errors. For example, line 258 - replace “maker” with marker. The authors are encouraged to proof-read thoroughly the text before resubmission.
- Please, provide a list of primers used for the RT PCR analysis.
Author Response

(The authors gave the same response as above.)
